# Carbon Ion Radiotherapy for Retroperitoneal Sarcoma: A Single-Institution Study

**DOI:** 10.3390/cancers17203395

**Published:** 2025-10-21

**Authors:** Reiko Imai, Tsukasa Yonemoto, Nobuhito Araki, Hirotoshi Takiyama, Hiroaki Ikawa, Shigeru Yamada, Hitoshi Ishikawa

**Affiliations:** 1QST Hospital, National Institute of Quantum Science and Technology,4-9-1, Anagawa, Inage, Chiba Shi 263-8555, Chiba, Japanishikawa.hitoshi@qst.go.jp (H.I.); 2Division of Orthopedic Surgery, Chiba Cancer Center, Chiba Shi 260-8717, Chiba, Japan; 3Department of Orthopedic Surgery, Ashiya Municipal Hospital, Ashiya Shi 659-8502, Hyogo, Japan

**Keywords:** carbon ion radiotherapy, charged particle therapy, retroperitoneal sarcoma, soft tissue sarcoma, radiation therapy

## Abstract

Retroperitoneal sarcoma (RPS) is a rare and challenging malignancy, particularly when surgical resection is not feasible due to anatomical complexity or patient-related factors. Carbon ion radiotherapy (CIRT), with its superior dose distribution and high linear energy transfer, offers a promising alternative for unresectable RPS. This single-institution retrospective study evaluated the clinical outcomes of 76 patients treated with CIRT between 2000 and 2022. The 3- and 5-year overall survival rates were 68.3% and 49.4%, respectively. The 3- and 5-year local control rates were 79.0% and 72.0%, respectively. Among 47 treatment-naïve patients, the 3- and 5-year abdominal recurrence-free survival rates were 51.1% and 29.1%, respectively. Late grade ≥3 adverse events occurred in 5.2% of patients. Compared to the STRASS trial (EORTC-62092), our cohort presented a more medically complex population, excluded from standard surgical approaches. These findings support CIRT as a definitive treatment option for unresectable RPS, warranting further multicenter studies to validate its efficacy and explore histology-specific therapeutic strategies.

## 1. Introduction

Retroperitoneal sarcomas (RPSs) are rare soft tissue tumors, accounting for approximately 15% of all soft tissue sarcomas and only 0.15% of all malignancies [1,2]. The mean annual incidence rate is 2.7 cases per 100,000 people and has remained significantly unchanged over time. The management of RPSs is primarily surgical resection, with 70.4% of patients undergoing resection [3]. RPS originates from the retroperitoneum in the upper abdomen and extends into the pelvic cavity. Surgical approaches vary by tumor location. Tumors occurring in the upper abdominal retroperitoneum may require en bloc organ resection. Meanwhile, tumors involving the iliac arteries in the pelvic cavity may necessitate hemipelvectomy. The 5-year survival rate of patients with RPSs ranges from 12% to 70% [4]. The role of radiotherapy (RT) as perioperative therapy remains controversial. Although several clinical trials have been conducted, none have yielded conclusive evidence demonstrating the effectiveness of perioperative RT [5,6]. The EORTC-62092 (STRASS) randomized phase 3 trial evaluated the impact of neoadjuvant RT on abdominal recurrence-free survival (ARFS) [6], but did not demonstrate a significant advantage of neoadjuvant RT.

Carbon ion radiotherapy (CIRT) for unresectable sarcomas has been available in Japan since 1996 [7]. CIRT utilizes carbon ion beams, which possess unique physical and biological properties. These beams feature a high linear energy transfer (LET) and a Bragg peak. These characteristics enable precise dose delivery to the tumor while minimizing exposure to surrounding normal tissues. Compared to protons, carbon ions are heavier, resulting in a lower penumbra and a more favorable dose distribution. Among clinically used particle beams, only carbon ions provide high LET, which is beneficial for treating radioresistant tumors such as sarcomas. Consequently, CIRT is expected to achieve superior local control (LC) in these malignancies. In 2018, Imai et al. reported the oncological outcomes of CIRT for soft tissue sarcomas [8]. In 2009, Serizawa et al. described its application in RPSs [9]. The present study aimed to provide an updated analysis of the clinical outcomes of CIRT for RPS at our institution.

## 2. Materials and Methods

### 2.1. Patients

This study retrospectively analyzed patients treated with CIRT for RPS between April 2000 and March 2022. All patients were enrolled in protocol studies, 9901(2) and 9901(3), investigating CIRT for sarcoma. The main eligibility criteria for these protocols were as follows: (1) an unresectable tumor as determined by surgeons or refusal of surgery by the patient; (2) histologically confirmed sarcoma; (3) a grossly measurable tumor; (4) absence of distant metastasis; (5) no prior RT to the tumor site, except in cases of radiation-associated sarcoma; and (6) provision of informed consent for treatment. The criteria for identifying RPS as defined in the STRASS trial were as follows: the tumor had to be unifocal, not extending through the sciatic notch or across the diaphragm, and not originating from bone structures, abdominal organs, or gynecological viscera [6].

### 2.2. Follow-Up and Statistical Analysis

The final follow-up was conducted in March 2024. Physical examination, computed tomography (CT) scans, and magnetic resonance imaging (MRI) were performed every 3–6 months during the first 5 years. Thereafter, follow-up was conducted every 6–12 months using CT and MRI. The examination interval was adjusted according to each patient’s condition. When regular follow-up at our institution was not feasible, the patients were referred to local hospitals. MRI and CT images obtained at the referring hospital were shared with us to facilitate the evaluation of treatment efficacy. Follow-up was maintained by directly contacting the patients via telephone and mailing questionnaires regarding tumor status.

The follow-up period was calculated from the initial date of CIRT. Recurrence was defined as tumor regrowth observed on two consecutive MRI or CT scans. When tumor regrowth was detected, an increased fluorodeoxyglucose (FDG) uptake on positron emission tomography-CT (PET-CT) compared with previous PET-CT images was used to support the diagnosis of local recurrence. Survival curves were generated from the initiation of CIRT using the Kaplan–Meier method. Late adverse events (AEs), defined as those occurring 90 days after treatment, were classified according to the Common Terminology Criteria for Adverse Events version 5.0 (US National Cancer Institute, Bethesda, MD, USA). Kaplan–Meier survival estimation was performed using Prism version 8.0 (GraphPad Software, San Diego, CA, USA). A 95% confidence interval (CI) was applied, and a *p*-value of <0.05 was considered statistically significant. Abdominal recurrence was defined as either local recurrence or intra-abdominal metastasis. In accordance with the STRASS trial criteria, liver metastases were classified as distant metastatic events rather than abdominal recurrences.

Informed consent was obtained from all patients. The study was conducted in accordance with the Declaration of Helsinki and approved by the Institutional Ethics Committee on Human Clinical Research of QST (N22-003, 20 June 2022).

### 2.3. Carbon Ion Radiotherapy

The relative biological effectiveness (RBE) dose was applied in CIRT. The prescribed dose of 70.4 Gy was delivered in 16 fractions over 4 weeks. In accordance with the protocol of that era, total doses of 64 Gy and 73.6 Gy were also employed. Respiratory gating was performed. Organs at risk (OARs) included the digestive organs and spinal cord. The kidney and ureter on the tumor site and the peripheral nerves were not considered OARs. The dose constraint for the intestine was set at a Dmax of <50 Gy. However, if the tumor was close to the intestine and the patient did not undergo prior abdominal surgery, Dmax < 60 Gy was applied. Okonogi et al. reported that a D2_cc_ of <56 Gy correlated with a lower rate of grade 1 proctitis development. This constraint was adopted for patients in the present study after its publication [10]. The spinal cord dose constraint was set at a Dmax of <40 Gy; however, when the tumor was in close proximity to the spinal cord, a Dmax of <50 Gy was permitted following patient counseling regarding the risk of radiation myelopathy. The minimum dose to 1 cc of the most irradiated spinal cord volume was maintained at <30 Gy [11]. Based on prior experience with CIRT for axial sarcoma, a total dose of 22.0 Gy delivered in 5 fractions to the entire spinal cord was considered safe. The dose constraint for the bladder was set at a Dmax of 63 Gy; all dose constraints were based on a schedule of 16 fractions over 4 weeks. The isocenter was defined as the target reference point dose. The treatment plans were designed to deliver >95% of the prescribed dose to the gross tumor volume (GTV) and >90% of the prescribed dose to the planning target volume (PTV). The GTV was delineated using CT (slice thickness: 2.0–5.0 mm) and MRI using contrast medium, and FDG PET-CT used as an adjunct when available, although it was not essential. Since 2013, the delivery method has gradually transitioned from passive scanning beams to active scanning beams. During the same period, the CT protocol for treatment planning was modified. The treatment planning CT scans were acquired with a slice thickness of 1–2 mm using respiratory gating. The clinical target volume (CTV) was defined as the GTV plus a margin of at least 1 cm in the tumor-invading tissue. The organ compartment and histology were considered when defining the CTV. The method described by Kawaguchi et al., which has been applied in orthopedic surgery, was referenced to determine the CTV for bone and soft tissue sarcomas [12]. In the passive setting, the PTV was defined as the CTV + 5 mm. In the active scanning setting, the PTV was determined based on the tumor shape and location, with margins ranging from 1 to 5 mm beyond the CTV. However, the treatment planning system (TPS) of CIRT is unique and different from commercial-based XRT TPS. It is difficult to adjust the OAR dose close to the tumor automatically. For example, when the tumor is extremely close to the spinal cord, complying with the dose constraint of the spinal cord can sometimes create a dose hotspot at the tumor edge. In such cases, reducing the PTV margin can sometimes help reduce the hotspot dose. If a steep change in tumor shape was observed between planning CT slices, a margin of 3–5 mm beyond the CTV was applied for the PTV. The PTV was not automatically generated. The dose constraints for OARs were prioritized in determining the dose distribution.

## 3. Results

### 3.1. All Patients

A total of 76 patients with RPSs were included in this analysis. Of these, 54 patients were followed until death, whereas 22 were alive at the time of the final follow-up. Among the 22 surviving patients, the intervals between the final follow-up and data collection for the present study were less than 1 year in 17 patients but more than 1 year in the remaining 5 patients. Of these five, two had follow-up periods longer than 10 years, and three had follow-up periods exceeding 40 months. The median follow-up duration for the entire cohort was 58.1 months (range, 2.3–212.8 months).

The median age at CIRT was 60.5 years (range, 14–83 years). The cohort included 44 men and 32 women. Disease status at the time of CIRT was as follows: 47 patients had been newly diagnosed with treatment-naïve tumors, 20 experienced recurrences after undergoing surgical resection, 2 had residual tumors post-resection, and 7 presented with solitary metastasis. The predominant histological subtype was dedifferentiated liposarcoma. The median PTV was 508.6 cm^3^ (range, 62–2872 cm^3^), and the median maximum diameter was 10 cm. The patient characteristics are summarized in Table 1.

In the entire cohort, the 3- and 5-year overall survival (OS) rates were 68.3% (95% CI, 56.5–77.5%) and 49.4% (95% CI, 36.7–60.9%), respectively. The median overall survival time was 58.1 months (range, 2.3–212.8 months). The 3- and 5-year LC rates were 79.0% (95% CI, 66.4–87.3%) and 72.0% (95% CI, 57.7–82.1%), respectively (Figure 1). Among the 56 patients who experienced treatment failure, the initial failure site was local recurrence in 10 cases (18%), abdominal recurrence excluding local recurrence in 20 cases (36%), and distant metastasis in 21 cases (38%). Simultaneous failures at multiple sites occurred in 5 cases (9%). The lung was the most common site of distant metastasis in 19 patients, including both cases with initial recurrence limited to the lung and those with simultaneous recurrence at multiple sites. The 3-year and 5-year ARFS rates were 50.2% (95% CI, 38.3–61.0%) and 33.8% (95% CI, 22.8–45.2%), respectively, with a median ARFS time of 39.2 months. The 3-year and 5-year disease-free survival (DFS) rates were 39% (95% CI, 27.9–49.7%) and 23% (95% CI, 13.7–32.9%), respectively.

Regarding the histology, dedifferentiated liposarcoma (DDLPS) accounted for 20% of all cases, undifferentiated pleomorphic sarcoma (UPS) for 18%, leiomyosarcoma for 11% and malignant peripheral nerve sheath tumor (MPNST) for 8%. Depending on the histology, 3-year overall survival rates were 66.7% (95% CI, 19.5–90.5%) in MPNST, 66.7% (95% CI, 37.5–84.6%) in UPS, 71.4% (95% CI, 40.6–88.2%) in DDLPS, 62.5% (95% CI, 22.9–86.1%) in leiomyosarcoma (Figure 2). The 3-year ARFS rates were 66.7% (95% CI, 19.5–90.5%) in MPNST, 60.0% (95% CI, 31.8–79.7%) in UPS, 38.5% (95% CI, 14.1–62.8%) in DDLPS, 50.0% (95% CI, 15.2–77.5%) in leiomyosarcoma, respectively (Figure 3). The 3-year LC rates were 66.7% (95% CI, 19.5–90.5%) in MPNST, 100% in UPS, 75.2% (95% CI, 40.7–91.4%) in DDLPS, 85.7% (95% CI, 33.4–97.9%) in leiomyosarcoma, respectively (Figure 4). No significant differences in OS, ARFS, or LC were observed among histological subtypes of RPSs using the Gehan-Breslow-Wilcoxon test and the Log-rank (Mantel–Cox) test.

### 3.2. Naïve Case

The characteristics of 47 patients with treatment-naïve tumor cases are summarized in Table 2. The 3- and 5-year OS rates were 65.6% (95% CI, 50.1–77.4%) and 45.3% (95% CI, 35.2–60.1%), respectively, with a median OS time of 57 months. At the time of evaluation, 12 patients remained alive. The 3- and 5-year ARFS rates were 51.1% (95% CI, 35.6–64.6%) and 29.1% (95% CI, 15.9–43.7%), respectively, with a median ARFS time of 39.2 months. The 3- and 5-year LC rates were 81.0% (95% CI, 63.8–90.6%) and 68.8% (95% CI, 48.3–82.5%), respectively. The 3- and 5-year disease-free survival rates were 41.4% (95% CI, 27.1–55.1%) and 17.9% (95% CI, 9.2–30.6%), respectively. (Figure 5). Among the 34 cases of treatment failure, the initial failure site was local recurrence in 7 cases (21%), abdominal recurrence excluding local recurrence in 8 cases (24%), and distant metastasis in 15 cases (44%). Simultaneous failures at multiple sites occurred in 4 cases (12%). The lung was the most common site of the first distant metastasis seen in 13 patients.

### 3.3. Adverse Events

None of the patients developed acute grade 3 AEs. Late grade 3 AEs occurred in four patients. One patient, who had normal ambulation prior to CIRT, required a wheelchair following treatment, which was classified as a grade 3 AE. With regard to gastrointestinal toxicity, one patient with left pelvic RPS underwent colostomy due to descending colon injury. Another patient with left pelvic RPS experienced a grade 5 AE resulting from small intestinal injury. This Grade 5 complication occurred in a patient with pelvic retroperitoneal sarcoma treated with CIRT after placement of a non-bioabsorbable spacer. Two months after CIRT, a severe infection developed around the spacer. We suspect that spacer displacement, as seen in the post-treatment CT scan, increased the radiation dose to the small intestine, leading to radiation enteritis and subsequent infection around the spacer. A grade 3 spinal bone fracture was observed in one patient. Although fixation surgery was recommended by orthopedic surgeons, the patient refused to undergo the procedure.

## 4. Discussion

Surgical resection remains the standard treatment for RPS. Several studies reported that microscopically negative margin resection (R0) was associated with improved OS compared to microscopically positive margin resection (R1) [13,14]. However, due to the complex anatomical location of RPS, achieving an R0 resection is often challenging. Institutional reports estimated that gross total resection rates ranged from 50% to 70% [15,16,17,18,19]. In other words, 30% to 50% of RPS cases did not achieve even gross total resection.

The candidates for CIRT typically comprise patients deemed medically unresectable by a multidisciplinary tumor board at a referral center. The reasons for unresectable tumors are summarized in Table 2. In total, 34% of patients had bone involvement, such as the spine and iliac bones, and 45% had arterial involvement, such as the aorta and common iliac artery. Tumor involvement of the external iliac artery requiring hemipelvectomy is considered highly invasive, particularly in elderly patients, and the appropriateness of surgical resection in such cases remains uncertain. Thus, medically unresectable cases encompass both patients having technically unresectable tumors and medically inoperable patients due to patient-related factors such as advanced age, comorbidities, or poor performance status. In this context, the 5-year OS rate of 49.4% in this study supports the clinical efficacy of CIRT for RPS.

The results of the EORTC-62092: STRASS were published in 2020 [6]. This is the first large-scale randomized study of RPS to evaluate the efficacy of preoperative RT on ARFS. In STRASS, the 3-year ARFS rates were 57.7% in the surgery group and 60.4% in the preoperative RT plus surgery group, indicating no significant benefit of preoperative RT on ARFS. In our study, the 3-year ARFS rate among naïve cases with baseline characteristics similar to those in STRASS was 51.1%, which seemed inferior to the outcomes reported in STRASS. However, several differences in patient background and tumor characteristics likely contributed to this discrepancy. One key factor is surgical margin status. In STRASS, 95% and 96% of patients in the surgery and RT plus surgery groups had macroscopically en block resection, respectively, indicating R1 or R0 status, because patients for whom surgery were expected to result in R2 resection based on the CT scans before randomization were excluded from STRASS candidates. Moreover, none of the patients in STRASS underwent hemipelvectomy or spinal body resection to achieve macroscopically complete resection. By contrast, patients in our study were judged to be medically unresectable; in some cases, hemipelvectomy was recommended to achieve complete resection. As shown in Table 2, 34% of the naïve cases exhibited bone invasion. Considering these factors, the majority of our patients were excluded based on the STRASS eligibility criteria. The candidates of our cohort are fundamentally different from those of STRASS. Another contributing factor is histology; only one (1.3%) patient in our cohort had well-differentiated liposarcoma. However, in STRASS, 32% of patients in the surgery group and 35% of those in the RT plus surgery group had well-differentiated liposarcoma, who are expected to have a long life expectancy. This could account for the observed differences in ARFS between the STRASS trial and our study.

In our cohort, with regard to histology, ARFS and OS rates were similar in MPNST, UPS, and leiomyosarcoma, suggesting that abdominal recurrence had a direct impact on OS in these subtypes. By contrast, DDLPS showed the highest OS rate, despite the lowest ARFS among the four histology groups. This finding indicates that abdominal recurrence did not necessarily lead to poor OS in the DDLPS. The higher 3-year LC rates observed in UPS and leiomyosarcoma may be attributed to a greater proportion of patients with a shorter overall prognosis compared with those with MPNST and DDLPS. These findings suggest that UPS and leiomyosarcoma may require more intensive systemic therapy to control abdominal recurrence outside the irradiation field. Meanwhile, in DDLPS, a different therapeutic approach may be necessary to control the disease, distinct from the strategies used for UPS and leiomyosarcoma.

RPS is sometimes located adjacent to the intestine, making high-dose irradiation challenging due to the intestinal dose constraints. Although carbon ion beams provide a superior dose distribution compared to X-rays, tumors in close contact with the intestine require a dose reduction to the adjacent area, which may increase the risk of recurrence. Previous studies have reported that the use of a surgical mesh, originally intended for abdominal wall hernia repair, creates a physical separation between the tumor and the intestine [20,21]. More recently, a bioabsorbable spacer has been investigated, which temporarily increases the distance between the tumor and the intestine, resulting in improved dose distribution to the tumor [22,23]. In the present study, the bioabsorbable spacer was surgically inserted prior to CIRT in four patients, allowing adequate dose delivery even to tumor regions adjacent to the intestine. There was no local recurrence in the four patients during a follow-up period ranging from 22 to 50 months.

In our study, 12 patients experienced inside irradiation field recurrence. Several factors appear to contribute to infield recurrence, one of which is the tumor’s inherent radiation sensitivity. In the 12 patients, five had DDLPS, and the remaining cases showed a variety of histological subtypes. Since soft tissue sarcoma is generally radioresistant, one strategy to improve LC using carbon ion beams involves escalating the total radiation dose delivered to the tumor. However, in previous clinical trials, we reported that a total dose of 73.6 Gy resulted in severe AEs in surrounding organs, highlighting the need for caution in dose escalation [7]. Another potential strategy involves the escalation of LET within the tumor. Matsumoto et al. reported that in G2 chondrosarcoma, a histologically homogeneous group, a low minimum dose-averaged LET (LETd) was associated with in-field local recurrence [24]. In pancreatic cancer, the LETd had a significant effect on local recurrence [25]. Okonogi et al. reported that LET was not associated with normal tissue toxicity [26].

The present study has several limitations. First, it was a retrospective analysis conducted at a single institution, which may introduce selection bias. Second, the cohort consisted of a relatively small number of patients with heterogeneous tumor characteristics and histological subtypes, which limited the generalizability of the findings. Third, detailed information regarding the use of chemotherapy was not available, which may have influenced treatment outcomes. Lastly, although long-term follow-up was achieved in most cases, the small number of patients in certain subgroups may have affected the statistical power of the analyses.

## 5. Conclusions

CIRT demonstrated favorable LC, OS, and ARFS outcomes in patients with medically unresectable RPSs. The population of our cohort is fundamentally different from those included in surgical trials such as STRASS. Our cohort represents a more challenging group for whom surgery was not feasible due to tumor location, comorbidities, or age. In this context, CIRT should not be viewed as inferior to surgery, but rather as a necessary and effective treatment option for patients excluded from standard surgical approaches. Histology-specific differences in recurrence and survival patterns further suggest the need for tailored therapeutic strategies in this population.

## Figures and Tables

**Figure 1 cancers-17-03395-f001:**
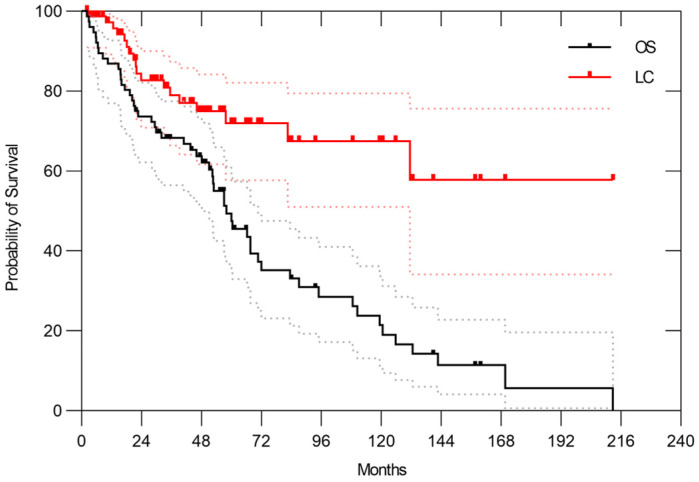
Overall survival and local control rates in patients with RPS treated with CIRT. In 76 patients, the 3- and 5-year OS rates were 68.3% (95% CI, 56.5–77.5%) and 49.4% (95% CI, 36.7–60.9%), respectively. Meanwhile, the 3- and 5-year LC rates were 79.0% (95% CI, 66.4–87.3%) and 72.0% (95% CI, 57.7–82.1%), respectively. The dotted lines represent the 95% confidence intervals.

**Figure 2 cancers-17-03395-f002:**
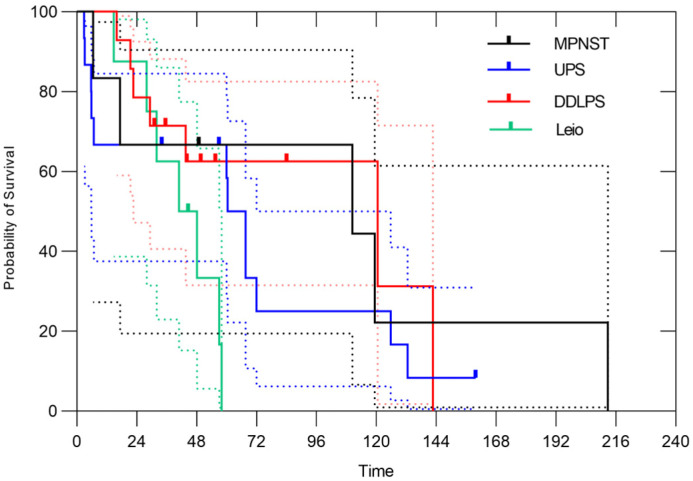
In all cases, the 3-year OS rates were 66.7% (95% CI, 19.5–90.5%) in MPNST, 66.7% (95% CI, 37.5–84.6%) in UPS, 71.4% (95% CI, 40.6–88.2%) in DDLPS, 62.5% (95% CI, 22.9–86.1%) in leiomyosarcoma, respectively. The dotted lines represent the 95% confidence intervals.

**Figure 3 cancers-17-03395-f003:**
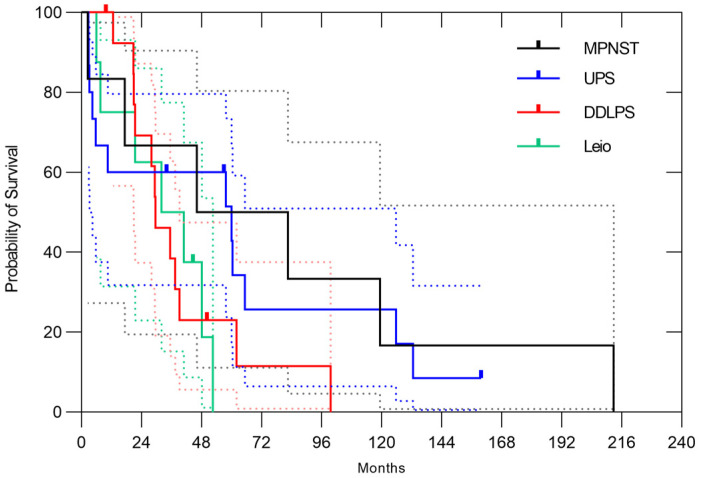
In all cases, the 3-year ARFS rates were 66.7% (95% CI, 19.5–90.5%) in MPNST, 60.0% (95% CI, 31.8–79.7%) in UPS, 38.5% (95% CI, 14.1–62.8%) in DDLPS, 50.0% (95% CI, 15.2–77.5%) in leiomyosarcoma, respectively. The dotted lines represent the 95% confidence intervals.

**Figure 4 cancers-17-03395-f004:**
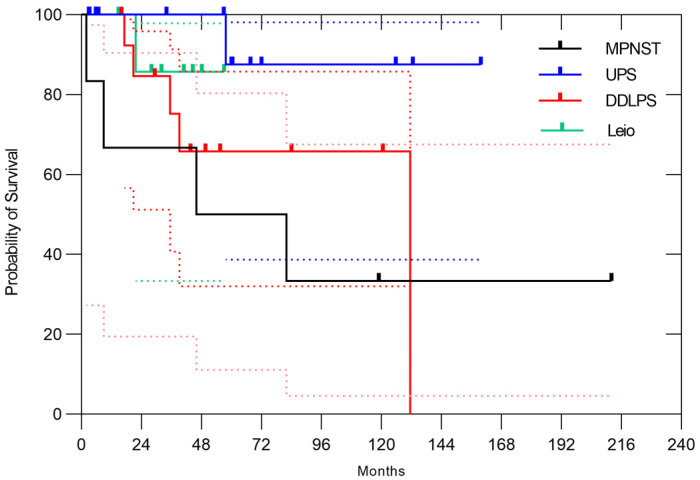
In all cases, the 3-year LC rates were 66.7% (95% CI, 19.5–90.5%) in MPNST, 100% in UPS, 75.2% (95% CI, 40.7–91.4%) in DDLPS, 85.7% (95% CI, 33.4–97.9%) in leiomyosarcoma, respectively. The dotted lines represent the 95% confidence intervals.

**Figure 5 cancers-17-03395-f005:**
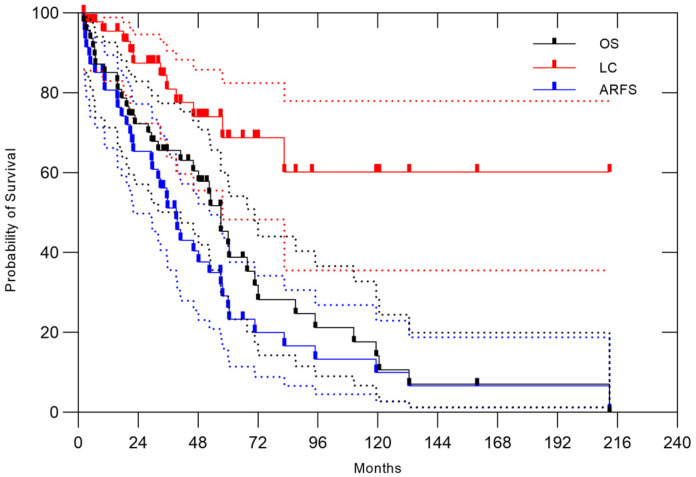
Overall survival, local control, and abdominal recurrence-free survival in 47 patients with treatment-naïve RPS treated with CIRT. The 3- and 5-year OS rates were 65.6% (95% CI, 50.1–77.4%) and 45.3% (95% CI, 35.2–60.1%, respectively; the 3- and 5-year ARFS rates were 50.2% (95% CI, 35.6–64.6%) and 29.1% (95% CI, 15.9–43.7%), respectively; and the 3- and 5-year LC rates were 81.0% (95% CI, 27.1–55.1%) and 17.9% (95% CI, 9.2–30.6%), respectively. The dotted lines represent the 95% confidence intervals.

**Table 1 cancers-17-03395-t001:** Patient characteristics.

	n (%)
Total number of patients	76 (100)
Median age (years)	61
Sex	
Male	44 (58)
Female	32 (42)
Tumor status	
Naïve	47 (62)
Post-operative recurrence	20 (26)
Residual tumor after resection	2 (2)
Solitary metastasis	7 (9)
Histology	
Liposarcoma	22 (29)
Well-differentiated	1
Myxoid	3
Pleomorphic	3
Dedifferentiated	15
MFH/UPS	14 (18)
Leiomyosarcoma	8 (11)
MPNST	6 (8)
Others	26 (34)
Location	
Upper abdomen	45 (59)
Pelvis	31 (41)
Total irradiation dose	
73.6	1 (1)
70.4	74 (98)
64.0	1 (1)
Median maximum diameter (cm)	10 (2.5–21)
Median PTV (cm^3^)	508.6

**Table 2 cancers-17-03395-t002:** Baseline characteristics of patients with treatment-naïve tumors.

	n (%)
Total number of patients	47 (100)
Median age (years)	61 (14–83)
Sex	
Male	25 (53)
Female	22 (47)
Histology	
Liposarcoma	7 (15)
Pleomorphic	1
Dedifferentiated	6
MFH/UPS	10 (21)
Leiomyosarcoma	5 (11)
MPNST	6 (12)
Others	19 (41)
Location	
Upper abdomen	32 (70)
Pelvis	15 (30)
Reasons for unresectability *	
Bone invasion	16
Arterial invasion	21
IVC invasion	4
Aging	7
Others	7
Median maximum diameter (cm)	11 (4–21)
Median PTV (cm^3^)	606 (67–2872)

* Including multiple reasons.

## Data Availability

The data supporting the findings of this study are available from the corresponding author upon reasonable request. Due to privacy and ethical restrictions, the datasets are not publicly available.

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
