# Peer review of "Carbon Ion Radiotherapy for Retroperitoneal Sarcoma: A Single-Institution Study"

_cancers, 2025, doi:10.3390/cancers17203395_

Round 1

Reviewer 1 Report

Comments and Suggestions for Authors

The work in question is certainly well written and very interesting, as it highlights an issue that is too often underestimated — namely, unresectable retroperitoneal sarcomas. The results of this analysis are certainly encouraging, thanks in part to the authors' expertise and experience in the use of carbon ion therapy. My only concern lies in the choice of inclusion criteria for the study.

In essence, I believe that if we aim to convey a clear and credible message, the core analysis of the work should focus on the 47 cases of previously untreated primary retroperitoneal sarcomas. Starting from such a homogeneous population, free from existing bias, would certainly give the results greater robustness and reproducibility.

The data concerning patients with recurrent or persistent disease, or with metastatic disease, involve clinical scenarios that are so heterogeneous that their outcomes have too many variables to standardize a treatment and demonstrate its efficacy

Author Response

Comment 1
The work in question is certainly well written and very interesting, as it highlights an issue that is too often underestimated — namely, unresectable retroperitoneal sarcomas. The results of this analysis are certainly encouraging, thanks in part to the authors' expertise and experience in the use of carbon ion therapy. My only concern lies in the choice of inclusion criteria for the study.

In essence, I believe that if we aim to convey a clear and credible message, the core analysis of the work should focus on the 47 cases of previously untreated primary retroperitoneal sarcomas. Starting from such a homogeneous population, free from existing bias, would certainly give the results greater robustness and reproducibility.

The data concerning patients with recurrent or persistent disease, or with metastatic disease, involve clinical scenarios that are so heterogeneous that their outcomes have too many variables to standardize a treatment and demonstrate its efficacy.

Thank you very much for your thoughtful and encouraging feedback. We appreciate your recognition of the clinical relevance of our work and the potential of carbon-ion radiotherapy (CIRT) for unresectable retroperitoneal sarcomas (RPSs).

We fully agree with your concern regarding the heterogeneity of the cohort. While we have not restructured the manuscript to focus exclusively on the naïve group, we have newly added details of the first failure sites in the naïve cases, in line with the analysis presented for all RPS cases. Additionally, we have clarified in the text that untreated cases are expected to benefit most from CIRT, given their clinical characteristics and the limitations of surgical options in this population.

Reviewer 2 Report

Comments and Suggestions for Authors

Carbon ion radiotherapy for retroperitoneal sarcoma: A single-institution study
Overview
CIRT is an effective definitive treatment for medically unresectable RPS, demonstrating favorable 3- and 5-year local control and overall survival rates with an acceptable safety profile. The study suggested CIRT is promising alternative to surgery for RPS patient’s ineligible for resection, though larger multicenter trials are needed to confirm these findings.
Major Comments
1. The 95% CI reported for the 3-year survival rates of different histological subtypes, 27.9% to 49.7% for multiple groups in Figures 2 to 4 are identical and narrow for the subgroup sample sizes. This raised a concern regarding an error in calculation, authors are requested to recheck the validity of the conclusions regarding histological differences.
2. Line 191: States ‘No significant differences... were observed’ among histological subtypes, but it does not present the results of any formal statistical tests, for examples ANOVA, to support this statement. Authors are requested to perform these analyses and include to justify the statement.
3. Line 142: PTV, a margin of 0 mm is unusual and raises concerns for underdosing the target. The rationale for this approach, especially the criteria for choosing a 0 mm margin, needs to be clearly stated.
4. Line 252: The comparison to the STRASS trial outcomes requires more depth. The discussion should more explicitly quantify the differences in patient populations, for examples, percentage with well-differentiated liposarcoma, expected R2 resection status. Authors are requested to acknowledge that the cohorts are fundamentally different, framing CIRT not as inferior but as a necessary option for a more challenging patient group excluded from STRASS.
5. While the rate of Grade 3+ adverse events is low, the single Grade 5 (Line 230) event (fatal small intestinal injury) requires more detailed discussion. Authors should elaborate on the circumstances of this event would provide context for evaluating the treatment’s risk profile.
Minor Comments
1. Please abbreviate at first mention. RPSs, CIRT are expanded and abbreviated few times. Please rectify.
2. The survival probability curves in Figures 2, 3, and 4 lack corresponding data points, tick marks, to indicate censored events, which is a standard feature of Kaplan-Meier plots.
3. While the standard dose is stated, a brief sentence explaining the rationale for the different total doses used (Line 114; 64.0 Gy and 73.6 Gy in one patient each) would be helpful.
4. Line 303: G3 AEs?
5. Line 310: Limitations should be stated clearly under separate section.
6. Conclusion should be rewritten, it should not provide a three-sentence summary.
Remark
The authors should address concerns raised and revise the manuscript accordingly. Please proofread the manuscript for typo and grammatical mistakes, professional help would be better.

Comments on the Quality of English Language

requires professional help

Author Response

Reviewer2

Major Comments
1. The 95% CI reported for the 3-year survival rates of different histological subtypes, 27.9% to 49.7% for multiple groups in Figures 2 to 4 are identical and narrow for the subgroup sample sizes. This raised a concern regarding an error in calculation; the authors are requested to recheck the validity of the conclusions regarding histological differences.

Thank you for your valuable comment. We have identified an error in the calculation of the 95% confidence intervals for the 3-year survival rates in Figures 2 to 4. The same CI range was mistakenly applied across subgroups with different sample sizes.

Correction:

Regarding the histology, dedifferentiated liposarcoma (DDLPS) accounted for 20% of all cases, undifferentiated pleomorphic sarcoma (UPS) for 18%, leiomyosarcoma for 11% and malignant peripheral nerve sheath tumor (MPNST) for 8%. Depending on the histology, 3-year overall survival rates were 66.7% (95% CI, 19.5 – 90.5%) in MPNST, 66.7% (95% CI, 37.5 – 84.6%) in UPS, 71.4% (95% CI, 40.6 – 88.2%) in DDLPS, 62.5% (95% CI, 22.9 – 86.1%) in leiomyosarcoma (Fig.2). The 3-year ARFS rates were 66.7% (95% CI, 90.5– 19.5%) in MPNST, 60.0% (95% CI, 31.8 – 79.7%) in UPS, 38.5% (95% CI, 14.1 – 62.8%) in DDLPS, 50.0% (95% CI, 15.2 – 77.5%) in leiomyosarcoma, respectively (Fig.3). The 3-year LC rates were 66.7% (95% CI, 19.5– 90.5%) in MPNST, 100% in UPS, 75.2% (95% CI, 40.7 – 91.4%) in DDLPS, 85.7% (95% CI, 33.4 – 97.9%) in leiomyosarcoma, respectively (Fig.4). No significant differences in OS, ARFS, or LC were observed among histological subtypes of RPSs using the Gehan-Breslow-Wilcoxon test and the Log-rank (Mantel–Cox) test.

We have revised the CIs to reflect accurate subgroup-specific estimates and updated the figures and related text accordingly. The conclusions regarding histological differences have been re-evaluated and remain valid.

We appreciate your feedback, which helped improve the accuracy of our analysis.

  1. Line 191: States ‘No significant differences... were observed’ among histological subtypes, but it does not present the results of any formal statistical tests, for examples ANOVA, to support this statement. Authors are requested to perform these analyses and include to justify the statement.

Thank you for your comment. We agree that a statistical test is needed to compare the OS, ARFS, and LC of four different histological types. We compared the curves using the Kaplan-Meier method. We used Prism 8.0, and there was no significant difference among the groups using the Gehan-Breslow-Wilcoxon test and the Log-rank (Mantel-Cox) test. Both tests showed no significant differences among the groups. We added it to the manuscript.

Correction:

No significant differences in OS, ARFS, or LC were observed among histological subtypes of RPSs using the Gehan-Breslow-Wilcoxon test and the Log-rank(Mantel-Cox)test.

  1. Line 142: PTV, a margin of 0 mm is unusual and raises concerns for underdosing the target. The rationale for this approach, especially the criteria for choosing a 0 mm margin, needs to be clearly stated.

Thank you for your comment. You are absolutely right — the margin should have been stated as 1 mm, not 0 mm. We apologize for the oversight. However, the TPS of CIRT is unique and different from commercial-based XRT TPS. It is difficult to adjust the OAR dose close to the tumor automatically. For example, when the tumor is extremely close to the spinal cord, complying with the dose constraint of the spinal cord can sometimes create a dose hotspot at the tumor edge. In such cases, reducing the PTV margin to 0mm can sometimes help reduce the hotspot dose. This approach is not standard and is only considered when the anatomical situation and dose constraints make it necessary.

Correction:

In the active scanning setting, the PTV was determined based on the tumor shape and location, with margins ranging from 1 to 5 mm beyond the CTV. However, the treatment planning system (TPS) of CIRT is unique and different from commercial-based XRT TPS. It is difficult to adjust the OAR dose close to the tumor automatically. For example, when the tumor is extremely close to the spinal cord, complying with the dose constraint of the spinal cord can sometimes create a dose hotspot at the tumor edge. In such cases, reducing the PTV margin can sometimes help reduce the hotspot dose.

  1. Line 252: The comparison to the STRASS trial outcomes requires more depth. The discussion should more explicitly quantify the differences in patient populations, for examples, percentage with well-differentiated liposarcoma, expected R2 resection status. Authors are requested to acknowledge that the cohorts are fundamentally different, framing CIRT not as inferior but as a necessary option for a more challenging patient group excluded from STRASS.

Thank you very much for your important comment. We agree that a more detailed comparison with the STRASS trial is important. We have already mentioned quantitative differences in patient characteristics, such as the proportion of well-differentiated liposarcoma and the expected resection status in the Discussion. We also explicitly acknowledge that the two cohorts are fundamentally different: STRASS enrolled resectable primary RPS patients, whereas our study focused on unresectable or surgery-refused cases. We have framed CIRT not as inferior but as a necessary treatment option for this more challenging patient group, which was excluded from STRASS.

  1. While the rate of Grade 3+ adverse events is low, the single Grade 5 (Line 230) event (fatal small intestinal injury) requires more detailed discussion. Authors should elaborate on the circumstances of this event would provide context for evaluating the treatment’s risk profile.

Thank you for your comment.

Correction:

This Grade 5 complication occurred in a patient with pelvic retroperitoneal sarcoma treated with CIRT after placement of a non-bioabsorbable spacer. Two months after CIRT, a severe infection developed around the spacer. We suspect that spacer displacement, as seen in the post-treatment CT scan, increased the radiation dose to the small intestine, leading to radiation enteritis and subsequent infection around the spacer.

Minor Comments
1. Please abbreviate at first mention. RPSs, CIRT are expanded and abbreviated few times. Please rectify.

Thank you for your comment. We agree with your point. We have revised the manuscript so that abbreviations like RPSs and CIRT are written in full at first mention, and then the short form is used after that.

  1. The survival probability curves in Figures 2, 3, and 4 lack corresponding data points, tick marks, to indicate censored events, which is a standard feature of Kaplan-Meier plots.

Thank you for your valuable observation regarding the Kaplan-Meier plots in Figures 2, 3, and 4. In response, we have revised the survival curves to include tick marks indicating censored events, in accordance with standard practice. The updated figures now clearly display these data points, and the figure legends have been modified to reflect this change.

  1. While the standard dose is stated, a brief sentence explaining the rationale for the different total doses used (Line 114; 64.0 Gy and 73.6 Gy in one patient each) would be helpful.

Thank you for your comment. The difference in dose is due to the different protocols used at the time of treatment. It just happened that there was one RPS case treated in each of those periods. Although the protocols were different, the schedule of four fractions per week for a total of 16 fractions was not changed.”

Correction:

 In accordance with the protocol of that era, total doses of 64 Gy and 73.6 Gy were also employed.

  1. Line 303: G3 AEs?

Thank you very much for your comment.

Correction:

We reported that a total dose of 73.6 Gy resulted in severe AEs

  1. Line 310: Limitations should be stated clearly under separate section.

Thank you very much for your valuable comment.

Correction:

This study has several limitations. First, it was a retrospective analysis conducted at a single institution, which may introduce selection bias. Second, the cohort consisted of a relatively small number of patients with heterogeneous tumor characteristics and histological subtypes, which limited the generalizability of the findings. Third, detailed information regarding the use of chemotherapy was not available, which may have influenced treatment outcomes. Lastly, although long-term follow-up was achieved in most cases, the small number of patients in certain subgroups may have affected the statistical power of the analyses.

  1. Conclusion should be rewritten, it should not provide a three-sentence summary.

Thank you very much for your valuable comment.

Correction:

 CIRT demonstrated favorable LC, OS, and ARFS outcomes in patients with medically unresectable RPSs. The population of our cohort is fundamentally different from those included in surgical trials such as STRASS. Our cohort represents a more challenging group for whom surgery was not feasible due to tumor location, comorbidities, or age. In this context, CIRT should not be viewed as inferior to surgery, but rather as a necessary and effective treatment option for patients excluded from standard surgical approaches. Histology-specific differences in recurrence and survival patterns further suggest the need for tailored therapeutic strategies in this population.